# The Potential Role of Human NME1 in Neuronal Differentiation of Porcine Mesenchymal Stem Cells: Application of NB-hNME1 as a Human NME1 Suppressor

**DOI:** 10.3390/ijms222212194

**Published:** 2021-11-11

**Authors:** Jin Hyoung Cho, Won Seok Ju, Sang Young Seo, Bo Hyun Kim, Ji-Su Kim, Jong-Geol Kim, Soon Ju Park, Young-Kug Choo

**Affiliations:** 1Department of Biological Science, College of Natural Sciences, Wonkwang University, 460, Iksan-daero, Iksan-si 54538, Korea; jojin83@hanmail.net (J.H.C.); jws7895@naver.com (W.S.J.); tjtkddud463@naver.com (S.Y.S.); jgkim84@wku.ac.kr (J.-G.K.); sjpark75@wku.ac.kr (S.J.P.); 2GreenBio Corp. Central Research, 201-19, Bubaljungand-ro, Bubal-eup, Icheon-si 17321, Korea; 3Institute for Glycoscience, Wonkwang University, 460, Iksan-daero, Iksan-si 54538, Korea; 4CHA Fertility Center Bundang, 59, Yatap-ro, Bundang-gu, Seongnam-si 13496, Korea; kimbh12@hanmail.net; 5Primate Resources Center (PRC), Korea Research Institute of Bioscience and Biotechnology, 181, Ipsin-gil, Jeongeup-si 56216, Korea; kimjs@kribb.re.kr

**Keywords:** macrophage, miniature pig adipose tissue-derived mesenchymal stem cells, nanobody, nucleoside diphosphate kinase A, ST8 alpha-N-acetyl-neuraminide alpha-2,8-sialyltransferase 1

## Abstract

This study aimed to investigate the effects of the human macrophage (MP) secretome in cellular xenograft rejection. The role of human nucleoside diphosphate kinase A (hNME1), from the secretome of MPs involved in the neuronal differentiation of miniature pig adipose tissue-derived mesenchymal stem cells (mp AD-MSCs), was evaluated by proteomic analysis. Herein, we first demonstrate that hNME1 strongly binds to porcine ST8 alpha-N-acetyl-neuraminide alpha-2,8-sialyltransferase 1 (pST8SIA1), which is a ganglioside GD3 synthase. When hNME1 binds with pST8SIA1, it induces degradation of pST8SIA1 in mp AD-MSCs, thereby inhibiting the expression of ganglioside GD3 followed by decreased neuronal differentiation of mp AD-MSCs. Therefore, we produced nanobodies (NBs) named NB-hNME1 that bind to hNME1 specifically, and the inhibitory effect of NB-hNME1 was evaluated for blocking the binding between hNME1 and pST8SIA1. Consequently, NB-hNME1 effectively blocked the binding of hNME1 to pST8SIA1, thereby recovering the expression of ganglioside GD3 and neuronal differentiation of mp AD-MSCs. Our findings suggest that mp AD-MSCs could be a potential candidate for use as an additive, such as an immunosuppressant, in stem cell transplantation.

## 1. Introduction

As human life expectancy increases, the number of patients with terminal organ failure, such as that due to chronic intractable neurological diseases, continues to increase. Organ transplantation or cell therapy (stem cell transplantation) is an effective treatment approach for terminal organ failure patients. However, for successful clinical transplants, the imbalance between the supply and demand for human organs must be resolved [1]. The continual shortage of human organs or cells is stimulating research in the field of xenotransplantation, and pigs are currently considered the most suitable potential source of organs or cells. Consequently, the scientific barrier has been somewhat resolved owing to the introduction of organ-source miniature pigs produced by genetic engineering and the availability and application of novel immunosuppressive agents [2,3]. Additionally, through various studies, transplant research has developed rapidly in recent years, and much progress has been achieved regarding xenotransplantation owing to systematic research on these scientific barriers [4].

As an example, the survival of short-term transplants has achieved tremendous success, with the one-year total survival rate exceeding 80% following numerous advances in immunosuppressants targeting adaptive immune cells [5]. Nonetheless, this success has not translated into long-term benefits because most transplants are lost over time. Moreover, the mechanisms of various immune cell responses in xenotransplantation are unknown, which remains a limiting factor for successful clinical transplants [6,7]. Thus, current immunotherapeutic strategies targeting adaptive immune cells display limited effectiveness in promoting long-term graft survival, and researchers have expressed doubt regarding the exclusive role of adaptive immune cells in mediating transplant rejection. In fact, emerging studies suggest that both adaptive and innate immune cells participate in transplant rejection [8,9,10]. Moreover, transplant rejection in immunosuppressed patients emphasizes the key role of MPs, natural killer cells, dendritic cells, and mast cells, which are innate immune cells involved in transplant rejection [11,12,13,14,15]. Among the limiting factors of clinical transplants, MPs in particular are phagocytic innate immune cells that play an important role in defending the host, and various studies have revealed the involvement of MPs in immune rejection of organ transplants [16,17,18,19].

MPs comprise tissue-resident MPs and recruited monocyte-derived MPs, both of which play a key role in innate immunity as crucial mediators of transplant immunopathology [20,21,22]. In fact, the accumulation of MPs is a very serious form of immunological rejection that occurs during stem cell transplantation, and it has been found that MPs are involved in both organ and cell transplantation [23]. Moreover, the accumulation of dense MPs has been confirmed in various histological studies of transplant rejection, and these MPs are known to have various functions, such as phagocytosis, antigen presentation, cytokine production, immune regulation, and tissue repair [24,25,26,27,28]. These research trends suggest that the role of MPs in the recent emergence of xenotransplantation rejection is an interesting topic in the field of transplant immunology. However, despite various studies on the relationship between MPs and transplant rejection, few have examined the function and role of secreted proteins, known as the secretome, from accumulated MPs in xenogeneic stem cell transplantation.

MP secretomes have diverse and highly complex roles within immune responses; thus, their various effects on transplant outcomes reflect the need to reassess the roles of the secretomes of various MP types in xenotransplantation [29,30,31,32,33]. Therefore, to better understand the role of MP secretomes in cellular xenograft rejection, here we have designed and utilized an in vitro mimic xenogeneic stem cell transplantation immune model to describe the interactions between human MP secretomes and the mechanisms of neuronal differentiation of miniature pig mesenchymal stem cells (mp-MSCs).

Meanwhile, gangliosides are sialic acid-containing glycosphingolipids that are most abundant in the nervous system [34,35,36] and are known to function in cell proliferation, adhesion, migration, apoptosis, and cell–cell and cell–substratum interactions [37,38,39,40,41]. In particular, previous studies have demonstrated that gangliosides play an important role in the neuronal differentiation of MSCs [35,42,43]. Interestingly, ganglioside GD3 is one of the b-series gangliosides, known to be significantly involved in the neurogenesis, and it is also considered to play an important role in the maintenance and proliferation of the neural stem cell system [44,45,46].

Specifically, this paper presents the roles of human MP secretomes and nucleoside diphosphate kinase A (NME1) within the neuronal differentiation of mp-MSCs. To demonstrate the negative effect of NME1 on the neuronal differentiation of mp-MSCs, we also discuss the fundamental mechanism of recombinant NBs, named NB-hNME1, as a suppressor of hNME1. Finally, we conclude by highlighting a new possibility that NB-hNME1 contributes to the neuronal differentiation of mp-MSCs by suppressing hNME1 and may potentially be used for immunotherapeutic strategies in xenogeneic stem cell transplantation.

## 2. Results

### 2.1. Effect of the MP Secretome on Changes in Ganglioside Expression and Neuronal Differentiation of Mp AD-MSCs

We designed and utilized an in vitro mimic xenogeneic stem cell transplantation immune model using mp AD-MSCs and U937 cells (Figure 1a). The mp AD-MSCs used in this experiment were selected at less than 6 passages, showed spindle-shaped morphology, and proliferated actively in culture (Appendix A). The mp AD-MSCs expressed MSC surface markers CD90 and CD144 [47,48,49] and lacked the expression of hematopoietic markers CD34 and CD45 (Appendix A) [50,51,52]. The mp AD-MSCs exhibited the highest neuronal differentiation efficiency (Appendix A) and the highest mRNA expression of neuronal markers microtubule-associated protein 2 (MAP2), neurofilament medium (NF-M), nestine, and glial fibrillary acidic protein (GFAP) at 3–6 h post-induction with neuronal differentiation medium (Appendix A).

U937 cells have been widely used as a model to investigate diverse biological processes related to monocyte and MP function [53]. Here, we used phorbol 12-myristate 13-acetate (PMA) to induce differentiation of human monocyte U937 cells into an MP-like phenotype, and the differentiated MPs showed expression of cluster of differentiation molecule 14 (CD14) and integrin alpha M (CD11b), which are MP surface markers (Appendix A). Because the secreted proteins mostly have low abundance when compared to high-abundance contaminating proteins derived from serum-containing culture media, the fetal bovine serum (FBS) proteins often mask the low-abundance secreted proteins, which makes it difficult to detect the secreted proteins using matrix-assisted laser desorption/ionization time-of-flight mass spectrometry (MALDI-TOF MS) and interpret the profiling data [54]. Therefore, analyzing secretomes in serum-free medium reduces the complexity of the proteome, leading to improved identification of secreted proteins [55]. However, the cells undergoing serum starvation could disturb cell metabolism and proliferation and may increase the risk of cell cytolysis [56]. Thus, serum starvation, which is not affected by cell proliferation, was carried out within 48 h to collect proteins released without serum interference (Figure 1b and Appendix A).

The effects of MPs or macrophage secretion medium (MSM) on the proliferation and neuronal differentiation of mp AD-MSCs was evaluated using co-cultures of mp AD-MSCs with MPs or MSM. The proliferation of mp AD-MSCs decreased significantly in the presence of MSM, but not with MPs (Figure 1c,d); additionally, MSM significantly reduced neuronal differentiation by more than 80% compared to the control (Figure 1e,f) and decreased neuronal marker gene expression of mp AD-MSCs (Appendix A).

Gangliosides (Figure 1g) are primarily synthesized in the endoplasmic reticulum and are further modified in the Golgi apparatus by sequential addition of carbohydrate moieties to an existing acceptor lipid molecule [57]. High-performance thin-layer chromatography (HPTLC) was performed to confirm whether MSM causes changes in ganglioside expression during neuronal differentiation of mp AD-MSCs. Figure 1h,i show that treating mp AD-MSCs with MSM inhibits the expression of alpha-N-acetyl-neuraminide alpha-2,8-sialyltransferase 1 (ST8SIA1) and ganglioside GD3. These results suggest that MSM reduces ganglioside GD3 expression in mp AD-MSCs and subsequently decreases the neuronal differentiation of mp AD-MSCs.

### 2.2. MALDI-TOF MS and Proteomic Analysis of the Secretome of MPs in MSM

The effect of MSM was analyzed using comparative sodium dodecyl sulfate-polyacrylamide gel electrophoresis (SDS–PAGE) coupled with MALDI-TOF MS followed by proteomics analysis to identify the components of the MP secretomes. Of the 26 bands marked with arrows, 17 bands were identified in MSM obtained from PMA-stimulated MPs for 48 h (Figure 2a). Among the 17 identified proteins, 7 proteins (band 1: MMP9, matrix metallopeptidase 9; band 6: MMP1, matrix metalloproteinase 1; band 7: ENO1, alpha-enolase; band 9: ENO1; band 15: SOD1, superoxide dismutase; band 16: NME1, nucleoside diphosphate kinase A; band 17: LYZ, lysozyme) showed a 20-fold increase in secretion compared to the control (PMA-stimulated MPs for 12 h) (Figure 2b).

Proteins control and execute most cellular processes, and their properties are modified and controlled through interactions with other proteins (protein–protein interactions) or biomolecules [58]. In particular, protein–protein binding plays a very important role in molecular function, as it can cause selective degradation of certain proteins and modulate protein function and cell signaling pathways [59].

Figure 2c shows that two clusters were identified among the 17 proteins using the database for annotation, visualization, and integrated discovery functional annotation clustering tool (cluster 1 data not shown). Eleven proteins among the MP secretomes (PKM, pyruvate kinase; HSPA8, heat shock 70 kDa protein 8; PRDX1, peroxiredoxin 1; ENO1; SOD1; NME1; HSPA1, heat shock 70 kDa protein 1; TPI1, triosephosphate isomerase 1; PGAM1, phosphoglycerate mutase 1; CTSD, cathepsin D; MMP9) have been identified to be involved in protein binding. Using information obtained from the database for annotation, visualization, and integrated discovery gene ontology database (http://david.abcc.ncifcrf.gov) and UniProt (http://www.uniprot.org), these proteins were clustered into 8 categories based on the biological processes in which they were involved, as follows: negative regulation of metabolic process (15%), phosphorus metabolic process (15%), negative regulation of cellular process (15%), single-organism biosynthetic process (12%), positive regulation of multicellular organismal process (12%), regulation of cell death (12%), hematopoietic or lymphoid organ development (11%), and embryo implantation (8%) (Figure 2d). Among the 4 proteins (MMP9, ENO1, SOD1, and NME1) corresponding to the “regulation of cell death” group, NME1 was confirmed to bind to ST8SIA1 (Figure 4a). Moreover, hNME1 is a well-documented metastasis suppressor gene with suppressor activity demonstrated across a wide spectrum of human cancers, including melanoma and carcinomas of the breast, stomach, and thyroid [60,61,62]. To date, however, no studies have examined the neuronal differentiation of mp AD-MSCs, especially the expression of ganglioside GD3, so here, we targeted hNME1 (Figure 2e) as a ganglioside GD3-regulating protein.

### 2.3. Production of Recombinant hNME1 to Verity the Effect of hNME1 on the Neuronal Differentiation of mp AD-MSCs

Before determining whether NME1 controls ganglioside GD3 of mp AD-MSCs, we quantitatively evaluated NME1 secreted by MPs and its effects on the proliferation and neuronal differentiation of mp AD-MSCs using recombinant hNME1 (rhNME1) proteins.

The amount of NME1 secreted by MPs increased from 24 h after PMA stimulation and reached the maximum after 48 h (Figure 3a). Correspondingly, Figure 3b (lower panel) shows that stimulation with PMA caused a significant time-dependent decrease in the level of NME1 cellular protein in MPs. In general, the NME gene includes naturally occurring read-through transcription between the neighboring NME1 and nucleoside diphosphate kinase B (NME2) genes [60,63,64]. Stimulation with PMA did not suppress the mRNA levels of NME2 but significantly decreased those of NME1 and nucleoside diphosphate kinase A/B in a time-dependent manner (Figure 3b, upper panel). These results indicate that the PMA-stimulated MPs arrest the gene expression of NME1 and release the internal NME1 protein externally. Figure 3c shows the quantitative analysis results of external (secreted) and internal NME1 in the MSM (24 h, 0.894 ± 0.018 ng/mL; 48 h, 3.992 ± 0.033 ng/mL; 72 h, 3.790 ± 0.108 ng/mL) and MPs (0 h, 4.642 ± 0.150 ng/mL; 24 h, 2.444 ± 0.106 ng/mL; 48 h, 1.685 ± 0.062 ng/mL; 72 h, 0.272 ± 0.032 ng/mL).

Next, rhNME1 protein was produced and purified (Figure 3d) to determine whether it influences the proliferation and neuronal differentiation of mp AD-MSCs. This influence was evaluated by co-cultures of mp AD-MSCs and rhNME1. Compared to the control, the addition of rhNME1 decreased mp AD-MSC proliferation by more than 50%, which was accompanied by morphological changes, and significantly reduced neuronal differentiation by more than 80% (Figure 3e,f).

### 2.4. Verification of Newly Identified Interactions between hNME1 and pST8SIA1

Previous results have indicated that the decrease in ganglioside GD3 reduced the neuronal differentiation of mp AD-MSCs and that this reduction was due to NME1. Therefore, to examine the molecular interaction (protein–protein binding) between hNME1 and ST8SIA1 of mp AD-MSCs, we conducted the following experiments.

Recombinant pST8SIA1 constructs (Figure 4b) used for bacterial expression were designed with an N-terminal glutathione S-transferase (GST) tag to facilitate the purification of soluble protein. Immunoblot (IB) analysis of the purified fusion GST protein showed reactivity to antibodies generated against the N-terminus of GST, full-length pST8SIA1 (X1, X2, and X3), and partial-length pST8SIA1 (P1, P2 and P3), thus confirming the identity of the 27, 67, 65, 58, 44, 39, and 35 kDa bands, respectively (Figure 4c). Next, we conducted a pull-down assay using GFP-fused full- or partial-length pST8SIA1 expressed in cell lysates with His-tagged recombinant human NME1 (His-rhNME1) beads. The full-length domain of pST8SIA1 (GST-X1, X2, and X3) was able to bind His-rhNME1 beads, whereas partial-length pST8SIA1 (GST-P1 and P3) failed to bind His-rhNME1 beads (Figure 4d). Notably, this indicates that porcine NME1 (pNME1) does not bind with pST8SIA1 (Figure 4e). These results demonstrate that the P2 domain of pST8SIA1 plays a crucial role in binding to hNME1 and that only hNME1, not pNME1, binds to pST8SIA1.

The type and order of amino acid sequences that constitute proteins are highly important factors determining protein–protein binding [65,66,67]. To date, no studies have examined binding between pST8SIA1 and hNME1 (Appendix A), which is first presented in this paper. Here, we aimed to determine why only hNMEI1, and not pNME1, binds to pST8SIA1 by comparing the differences in amino acid sequences between hNME1 and pNME1.

Of the full amino acid sequences of hNME1 and pNME1, it was confirmed that eight amino acids differed. Furthermore, we manufactured recombinant pNME1 (rpNME1) mutants in which the eight amino acids from the porcine sequence were replaced with those from the human sequence (Figure 4f,g). Surprisingly, only mutant 7 (porcine lysine (K) was replaced with threonine (T) at the 143rd amino acid of hNME1), and mutant 8 (porcine alanine (A) was replaced with asparagine (N) at the 148th amino acid of hNME1) were confirmed to bind to pST8SIA1. Appendix A shows the structural formula of the four amino acids. Moreover, mutant 7+8 (in which the 143rd and 148th amino acids of hNME1 were replaced with T and N, respectively) was confirmed to bind more strongly to pST8SIA1 than mutants 7 or 8 individually (Figure 4h,i).

### 2.5. Correlation Analysis of the Neuronal Differentiation of mp AD-MSCs with Degradation of pST8SIA1 by hNME1

Previous studies have reported that specific proteins, when combined with other proteins, cause structural and functional disorders in various proteins [68,69,70,71,72]. As mentioned previously, ST8SIA1 is located in the Golgi apparatus of mp AD-MSCs, and the externally administered hNME1 was combined with ST8SIA1 of mp AD-MSCs. Treatment with rhNME1 also inhibited ST8SIA1 expression and caused morphological changes in mp AD-MSCs (Figure 5a); furthermore, while neither rhNME1 nor rpNME1 suppressed ST8SIA1 mRNA levels, rhNME1 significantly decreased the protein levels of ST8SIA1 in a dose-dependent manner (Figure 5b). In addition, when cycloheximide-pretreated mp AD-MSCs were incubated with rhNME1, MG132, or a combination of the two, ST8SIA1 protein degradation was enhanced (Figure 5c). These results indicate that externally administered rhNME1 was combined with ST8SIA1 and subsequently caused the degradation of ST8SIA1 protein in mp AD-MSCs.

Next, a knockdown of ST8SIA1 was performed to determine whether the reduction in ganglioside GD3 expression caused by ST8SIA1 suppresses the neuronal differentiation of mp AD-MSCs. We designed five pST8SIA1-specific small interfering RNAs (siRNAs) targeting regions #1, 2, 3, 4, and 5 of the pST8SIA1 genome (Appendix A). Ganglioside GD3 and both mRNA and protein levels of ST8SIA1 were most suppressed by porcine ST8SIA1 siRNA (siST8SIA1) #4 (si#4) in mp AD-MSCs (Figure 5d), and knockdown of ST8SIA1 significantly reduced the neuronal differentiation of mp AD-MSCs by more than 80% compared to the control (Figure 5e,f).

### 2.6. Production of NB-hNME1 for Inhibiting the Binding Capacity of hNME1 with pST8SIA1

Previous results have demonstrated that the decrease in ganglioside GD3, which was due to hNME1 binding with pST8SIA1, reduced the neuronal differentiation of mp AD-MSCs. Therefore, we focused on blocking the molecular interaction between hNME1 and pST8SIA1 in mp AD-MSCs, and the following procedure was carried out.

In order to produce NBs that block only hNME1 and not pNME1, we commissioned the production of NBs, named NB-hNME1, that only combine with hNMEI1, from Elpis Biotechnology (ELPIS-Biotech Inc., Daejeon, Korea). The hNME1 antigen-specific antibodies were detected by phage enzyme-linked immunosorbent assay (ELISA) (Figure 6a). NB-hNME1, which attached to both the T^143^ and N^148^ nearby domains in hNME1, was selected by hNME1 antigen epitope mapping (Figure 6d,e and Appendix A). NB-hNME1 was determined by antibody titer (Figure 6b), and the specificity of the hNME1 antigen in NB-hNME1 was confirmed by ELISA and IB analysis. Figure 6c shows that NB-hNME1 only binds to hNME1 and not pNME1; however, as an exception, NB-hNME1 also binds to pNME1 mutants 7, 8, and 7+8.

These results indicate that NB-hNME1 can effectively block the T^143^ and N^148^ domains of hNME1. Therefore, to determine whether NB-hNME1 interferes with hNME1 in mp AD-MSCs, further experiments were conducted.

### 2.7. Verification of the Recovery of Neuronal Differentiation of mp AD-MSCs by Treatment with NB-hNME1 as an hNME1 Suppressor

The mp AD-MSCs were treated with NB-hNME1 (5–20 µg/mL) for 72 h, and morphological changes were not observed within this concentration range (Figure 7a). However, significant cytotoxicity was observed from 24 to 72 h in mp AD-MSCs treated with NB-hNME1 (20 µg/mL) (Figure 7b). Therefore, we began with a non-significant concentration of NB-hNME1 (5 µg/mL) and increased the concentration 2- and 3-fold (10 and 15 µg/mL) for analysis in this study.

We used a GST-X1/His-rhNME1 pull-down assay to determine how increases in the concentration of NB-hNME1 affect the formation of the GST-X1/His-rhNME1 complex. The binding of GST-X1 and His-rhNME1 was significantly reduced by the addition of NB-hNME1 in a dose-dependent manner (Figure 7c). Furthermore, HPTLC and Western blot analysis showed that the addition of NB-hNME1 recovered the ganglioside GD3 expression levels that were reduced following co-culture with MSM or hNME1 in mp AD-MSCs (Figure 7d,e). Most importantly, immunocytochemistry confirmed that neuronal differentiation of mp AD-MSCs, which was reduced by rhNME1, was recovered by the addition of NB-hNME1 (Figure 7f). Phase-contrast images and cell counts of the neuronal differentiation of mp AD-MSCs also revealed that the ratio of NI-mp AD MSCs was recovered by NB-hNME1 (Figure 7g,h).

These results show that the reduction in ganglioside GD3 by hNME1 inhibits the neuronal differentiation of mp AD-MSCs and that these effects are ameliorated by the application of NB-hNME1.

## 3. Discussion

The therapeutic antibody has become one of the fastest-growing medicines since the first antibody treatment was developed and commercialized in 1994, and it is used for various purposes, such as anti-cancer and anti-inflammatory effects, autoimmune diseases, and transplant rejection [73,74,75]. Among therapeutic antibodies, mouse monoclonal antibodies are very useful for diagnostic reagents and basic research because they have a wide range of target antigens and can be mass produced; however, repeated administration to the human body for the purpose of treating diseases causes an immune response (HAMA, human anti-mouse antibody, response), resulting in side effects and reduced effectiveness, thus rendering them unusable as immunotherapy drugs [76].

To solve this problem, advanced countries have continuously developed technology for human antibody manufacturing to produce mouse antibodies similar to those of humans since 1988; furthermore, new human monoclonal antibody manufacturing technology has undergone development since 1990, and several antibodies have begun clinical trials [77]. For therapeutic antibodies, human monoclonal antibodies are the most desirable, but development has been delayed due to the difficulty of production using human hybridization technology. Recently, however, two techniques utilizing phage display and transgenic mice have been developed that can manufacture human monoclonal antibodies without relying on human hybridization technology [78,79]. However, reagents based on IgG antibodies produced through these two technologies exhibit some practical shortcomings [80].

Currently, with the emergence of antibody engineering, many problems have been overcome with the development of recombinant antibody fragments, such as Fab or scFv (single-chain antibody) and sdAb or NB (single-domain antibody); in particular, these fragments not only retain the specificity of the whole monoclonal antibodies but are also easy to express and produce in prokaryotic expression systems [81,82]. Therefore, we produced new target recombinant NBs, “NB-hNME1,” to suppress cellular immune rejection by human MPs occurring during xenogeneic stem cell transplantation, and the indirect effect of NB-hNME1 as an immunosuppressive targeted treatment additive was demonstrated in vitro.

We designed and utilized the in vitro mimic xenogeneic stem cell transplantation immune model using mp AD-MSCs and U937 cells. The proliferation of mp AD-MSCs decreased significantly in the presence of MSM, which significantly reduced the neuronal differentiation of mp AD-MSCs. These results indicate that MSM reduces the ganglioside GD3 expression of mp AD-MSCs and subsequently decreased the neuronal differentiation of mp AD-MSCs (Figure 1, and Appendix A).

More specifically, this study first demonstrated that hNME1, which is present in MSM, plays a crucial role in binding to the P2 domain of pST8SIA1. Furthermore, when hNME1 binds with pST8SIA1, it induces degradation of pST8SIA1 in mp AD-MSCs, thereby inhibiting ganglioside GD3 expression and subsequently decreasing the neuronal differentiation of mp AD-MSCs (Figure 2, Figure 3 and Figure 4, and Appendix A). Therefore, in order to block only hNME1 and not pNME1, we produced NB-hNME1 to bind specifically with hNME1, and the blocking effect of NB-hNME1 as an hNME1 suppressor was evaluated for its effect on binding between hNME1 and pST8SIA1. Remarkably, NB-hNME1 effectively blocked the binding of hNME1 to pST8SIA1, thereby recovering the expression of ganglioside GD3 and neuronal differentiation of mp AD-MSCs (Figure 5, Figure 6 and Figure 7, and Appendix A).

## 4. Materials and Methods

### 4.1. Cultivation and Characterization of mp AD-MSCs

The mp AD-MSCs were obtained from Korea Research Institute of Bioscience and Biotechnology (KRIBB, Daejeon, Korea). The mp AD-MSCs were cultured in standard culture medium comprising Dulbecco’s modified Eagle’s medium (DMEM; WelGene, Gyeongsan, Korea) supplemented with 10% FBS (Gibco, Gaithersburg, MD, USA), 10 ng/mL basic fibroblast growth factor (R&D systems, Minneapolis, MN, USA), and 100 U/mL Gibco penicillin-streptomycin (Gibco, Gaithersburg, MD, USA). The cells were cultured at 37 °C under 5% CO_2_ in a humidified chamber until passage 6 in the experiments. The morphological properties of the cultured mp AD-MSCs were observed regularly with an inverted phase-contrast microscope, and images were acquired with digital imaging software 6.1 (Carl Zeiss, Ulm, Germany). The mp AD-MSCs were characterized using the following conjugated monoclonal antibody combinations: CD_90_-PE/CD_44_-APC/CD_45_-PE and CD_34_-FITC. All antibodies were purchased from BD Biosciences (San Jose, CA, USA). The mp AD-MSCs (1 × 10^6^ cells) were suspended in Dulbecco’s phosphate-buffered saline (DPBS; WelGene, Gyeongsan, Korea) and incubated with combinations of the monoclonal antibodies described above, and untreated mp AD-MSCs were used as the control. Labeled cells were acquired immediately after staining using a FACS Calibur flow cytometer (BD Biosciences, San Jose, CA, USA).

### 4.2. Neuronal Differentiation of mp AD-MSCs

The in vitro neuronal differentiation of mp AD-MSCs was performed as previously described [83,84] with the following modifications. At 70–80% confluence, the growth medium was removed, and the mp AD-MSCs were induced with NI media comprising DMEM, 20% FBS, 1 mM β-mercaptoethanol (Sigma-Aldrich, St. Louis, MO, USA), and 50 ng/mL basic fibroblast growth factor for 24 h. After pre-induction, the NI medium was removed, and the mp AD-MSCs were washed with DPBS. To induce neuronal differentiation, the medium was replaced with neuronal differentiation medium comprising DMEM, 0.5% FBS, 2% dimethyl sulfoxide (Sigma-Aldrich, St. Louis, MO, USA), 200 μM butyrated hydroxyanisole (Sigma-Aldrich, St. Louis, MO, USA), 2 mM valproic acid (Sigma-Aldrich, St. Louis, MO, USA), 10 μM forskolin (Tocris Bioscience, Bristol, UK), 1 μM hydrocortisone (Tocris Bioscience, Bristol, UK), 5 μg/mL insulin (Thermo Fisher Scientific, Frederick, MD, USA), and 25 mM potassium chloride (Sigma-Aldrich, St. Louis, MO, USA). The mRNA and protein expression levels of neuronal marker genes [85,86,87] were determined by immunocytochemistry or reverse transcription polymerase chain reaction (RT-PCR). Immunocytochemistry was performed for MAP2, a mature neuronal marker. RT-PCR was performed for the late-stage neuronal markers MAP2 and NF-M and for the early-stage neuronal markers nestin and GFAP.

### 4.3. Morphological Evaluation and PMA-Induced Differentiation of U937

Human monocytic cell line U937 cells were purchased from Korean Cell Line Bank (KCLB; no. 21593.1, Seoul, Korea) and maintained in DMEM with 10% FBS and 100 U/mL penicillin–streptomycin. U937 cells were maintained at 37 °C in a humidified 5% CO_2_ atmosphere. To differentiate U937 into MPs, U937 cells were incubated for 48 h with 100 ng/mL PMA (Sigma-Aldrich, St. Louis, MO, USA) [53,88,89]. Adherent cells were selected as mature MPs, and cell culture media, including the secretory products of MPs, were used for further studies. The morphology of mature MPs was evaluated using a light microscope (Carl Zeiss, Ulm, Germany) after staining according to the manufacturer’s instructions with Wright-Giemsa solution (Polysciences, Inc., Warrington, PA, USA). MP surface marker genes, such as cluster of differentiation molecule 14 (CD14) [90] and CD11b, [91] were assessed by RT-PCR, IB analysis, and immunocytochemistry.

### 4.4. RT-PCR

The cDNA for RT-PCR was obtained by PCR amplification using primers specific to β-Actin, MAP2, NF-M, nestin, GFAP, CD14, CD11b, NME1, NME2, and nucleoside diphosphate kinase A/B. RT-PCR was performed using TaKaRa Ex Taq (Takara, Dalian, China) with the TaKaRa PCR Thermal Cycler Dice Gradient system (Takara, Dalian, China). Primer information and PCR conditions used in the study are provided in Appendix A.

### 4.5. IB Analyses

Proteins samples were separated by 9%, 12%, or 13.5% SDS-PAGE and then transferred to membranes, which were incubated overnight at 4 °C with specific antibodies. The protein bands were observed after incubation with horseradish peroxidase-conjugated secondary antibodies, using a SuperSignal West Pico PLUS chemiluminescent substrate (Thermo Fisher Scientific, Frederick, MD, USA).

### 4.6. Immunocytochemistry

Cells were rinsed in DPBS and fixed in 4% paraformaldehyde (Sigma-Aldrich, St. Louis, MO, USA) for 30 min. Cells were permeabilized by washing three times in DPBS containing 0.1% Triton X-100 (Sigma-Aldrich, St. Louis, MO, USA) and then blocked in DPBS containing 0.5% bovine serum albumin (Sigma-Aldrich, St. Louis, MO, USA) for 1 h at room temperature. Primary antibodies were added and incubated at 4 °C overnight. Probed cells were reacted with Alexa-Fluor secondary antibodies (Invitrogen, Carlsbad, CA, USA) and visualized using a fluorescence microscope (Carl Zeiss, Ulm, Germany).

### 4.7. Ganglioside Extraction and Purification

Gangliosides were prepared as previously described [92]. Total lipids were extracted with chloroform/methanol (1:1, *v*/*v*). Subsequently, neutral lipids were filtered off with 20 mL chloroform/methanol/H_2_O (15:30:4, *v*/*v*/*v*) by applying a DEAE Sephadex A25 column (Sigma-Aldrich, St. Louis, MO, USA), and then acidic lipids were extracted with 15 mL chloroform/methanol/0.8 M sodium acetate (15:30:4, *v*/*v*/*v*). The eluted samples were dried with N_2_ gas at 30 °C, dissolved in chloroform/methanol (1:1, *v*/*v*), and neutralized with 12N NH_4_OH overnight at room temperature. After drying neutralized samples again with N_2_ gas at 30 °C, dried samples were dissolved in distilled water, and the salt was removed with a Sep-Pak C18 cartridge (MilliporeSigma, Madison, WI, USA) to obtain gangliosides. Finally, eluted gangliosides were dried with N_2_ gas at 30 °C for 4 h. Dried samples were stored at −80 °C until further use.

### 4.8. HPTLC

HTPLC analysis was performed as previously described [92]. Purified gangliosides in chloroform/methanol (1:1, *v*/*v*) were run on HPTLC plates, which were developed with chloroform/methanol/0.25% CaCl_2_·H_2_O (50:40:10, *v*/*v*/*v*). The developed gangliosides were stained with resorcinol solution (HCl, 0.1M CuSO_4_·5H_2_O, resorcinol, distilled water). A monosialoganglioside mixture (Matreya LLC, State College, PA, USA) and disialoganglioside mixture (Matreya LLC, State College, PA, USA) were used as standard markers for individual ganglioside species.

### 4.9. Protein Identification by Mass Spectrometry

For protein identification by peptide mass fingerprinting (PMF), protein spots were excised, digested with trypsin (Promega, Madison, WI, USA), mixed with α-cyano-4-hydroxycinnamic acid in 50% acetonitrile/0.1% TFA, and subjected to MALDI-TOF MS analysis (Microflex LRF 20, Bruker Daltonics, Billerica, MA, USA), as described by Fernandez et al. [93]. Spectra were collected from 300 shots per spectrum over m/z range 600–3000 and calibrated by two-point internal calibration using trypsin auto-digestion peaks (*m*/*z* 842.5099, 2211.1046). The peak list was generated using Flex Analysis 3.0. Peak-picking thresholds were as follows: 500 for minimum resolution of monoisotopic mass, 5 for S/N. The search program MASCOT, developed by Matrixscience (http://www.matrixscience.com/), was used for protein identification by PMF. The following parameters were used for the database search: trypsin as the cleaving enzyme, a maximum of one missed cleavage, iodoacetamide (Cys) as a complete modification, oxidation (Met) as a partial modification, monoisotopic masses, and a mass tolerance of ±0.1 Da. The PMF acceptance criteria used probability scoring.

### 4.10. ELISA

Cell lysates and conditioned media from the mature MP assays were analyzed for hNME1 by ELISA. An hNME1 ELISA kit was purchased from FineTest (Wuhan Fine Biological Technology Co., Ltd., Hubei, China) and used as per the manufacturer’s instructions. Microplates were read on an Epoch 2 Microplate spectrophotometer (BioTek Instruments, Inc., Winooski, VT, USA). Cytokine levels were then extrapolated from a standard curve.

### 4.11. Expression Constructs

Wild-type (WT) pST8SIA, ganglioside GD3 synthetase, mutant cDNAs, and domains thereof were cloned along with a N-terminal GST tag into the pEX-N-GST vector (OriGene, Rockville, MD, USA) using the Hind III and Xho I restriction sites. Wild-type hNME1, pNME1, and mutant cDNAs were customized from Bio Basic Inc. (Markham, ON, Canada) and subcloned with a His-tag into BamH I/EcoR I-digested pET-28B(+) vector (MilliporeSigma, Madison, WI, USA). All final cDNA constructs were verified by DNA sequencing.

### 4.12. Transfection with siRNA

For transfection, siST8SIA1 was obtained from Genolution (Genolution Inc., Seoul, Korea). To transfect mp AD-MSCs with siRNA, cells were seeded in 96-well plates (10^4^/well), and Lipofectamine 2000 (Invitrogen, Carlsbad, CA, USA) with siRNA was added according to the manufacturer’s instructions. The cells transfected with scrambled siRNA (Genolution Inc., Seoul, Korea) were used as controls. The siST8SIA1 targeting and control sequences are listed in Appendix A.

### 4.13. Screening Peptide Microarray

A peptide microarray copy was pre-stained with secondary and control antibodies in incubation buffer to investigate background interactions with the antigen-derived peptides that could interfere with the main assay. Subsequently, other peptide microarray copies were incubated with NB-hNME1 at concentrations of 1, 10, and 100 µg/mL in incubation buffer followed by staining with secondary and control antibodies. Read-out was performed with a LI-COR Odyssey Imaging System at scanning intensities of 7/7 (red/green). The additional HA control peptides framing the peptide microarray were simultaneously stained with the control antibody as an internal quality control to confirm the assay quality and the peptide microarray integrity.

### 4.14. Peptide Microarray Spot Quantification

Quantification of spot intensities and peptide annotation were based on 16-bit gray scale tiff files, which exhibit a higher dynamic range than the 24-bit colorized tiff files. Microarray images were analyzed using a PepSlide Analyzer 2.0 (PEPperPRINT, Heidelberg, Germany). A software algorithm breaks down the fluorescence intensities of each spot into raw, foreground, and background signals and then calculates the averaged median foreground intensities and spot-to-spot deviations of spot duplicates. Based on averaged median foreground intensities, an intensity map was generated, and interactions in the peptide map were highlighted according to an intensity color code, with red indicating high and white representing low spot intensities. The tolerance of maximum spot-to-spot deviation was 40%; otherwise, the corresponding intensity value was zeroed. The averaged spot intensities of the assays were plotted with NB-hNME1 against the antigen sequence from the N- to the C-terminus to visualize overall spot intensities and signal-to-noise ratios. The intensity plots were correlated with peptide and intensity maps as well as with visual inspection of the microarray scans to identify the epitope of NB-hNME1.

### 4.15. Statistical Analysis

Results were presented as mean ± SD of at least 3 independent experiments. Data were analyzed for statistical significance using one-way analysis of variance. A *p* value < 0.05 was considered significant.

## 5. Conclusions

Based on a comprehensive evaluation of the experimental results, our findings suggest that NB-hNME1 can be used as an hNME1 suppressor for the neuronal differentiation of mp AD-MSCs and thus be a potential candidate for use as an additive, such as an immunosuppressant, in xenogeneic stem cell transplantation.

## Figures and Tables

**Figure 1 ijms-22-12194-f001:**
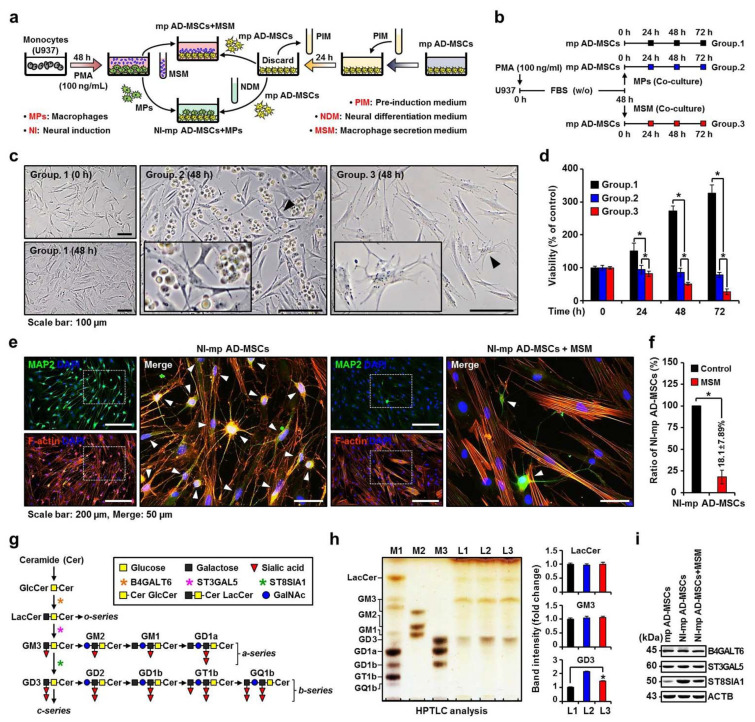
Expression patterns of gangliosides in the in vitro xenogeneic stem cell transplantation immune model. (**a**) The experimental setup to mimic xenogeneic stem cell transplantation in vitro. Human U937 cells were either used as monocytes or differentiated into MPs by incubation with 100 ng/mL PMA. (**b**) We used MPs and MSM, which were collected for 48 h in serum-free medium. Groups 1, 2, and 3 indicate the control, MP co-culture, and MSM co-culture, respectively. In a direct co-culture system, MPs were directly added to mp AD-MSC cultures. (**c**) Phase-contrast images of the co-culture system between mp AD-MSCs with MPs or MSM for 48 h. Arrowheads indicate enlarged images. (**d**) Cell proliferation was quantified by an MTS assay at 24, 48, and 72 h post-co-culture. Data are presented as mean percentage levels ± SD (*n* = 3; * *p* < 0.05). (**e**) Immunofluorescence staining of neuronal induction (NI)-mp AD-MSC culture (control) and co-culture with MSM using MAP2-FITC, F-actin, and DAPI. White squares indicate enlarged merged images; arrowheads indicate induced neural-like cells. (**f**) Variations in the cell count of NI-mp AD-MSCs were quantified for co-cultures with or without MSM after 48 h. Data are presented as mean percentage levels ± SD (*n* = 3; * *p* < 0.05). (**g**) General scheme for ganglioside synthesis; steps are also reversible. G: ganglioside; M: monosialo; D: disialo; numbers denote carbohydrate sequence. GlcCer: glucosylceramide; LacCer: lactosylceramide; GalNAc: N-acetylgalactosamine; B4GALT6, beta-1,4-galactosyltransferase 6; ST3GAL5, ST3 beta-galactoside alpha-2,3-sialyltransferase 5; ST8SIA1, ST8 alpha-N-acetyl-neuraminide alpha-2,8-sialyltransferase 1. (**h**) HPTLC of gangliosides in mp AD-MSCs (L1), NI-mp AD-MSCs (L2), or co-cultures with MSM (L3). M, marker; L, line; M1, adult mouse brain gangliosides; M2 and M3, ganglioside standard markers (left panel); quantification of band intensity of LacCer, GM3, and GD3 (right panel). Data are presented as fold changes ± SD (*n* = 3; * *p* < 0.05). (**i**) Western blots for gangliosides LacCer, GM3, and GD3 synthase. Expression patterns of B4GALT6, ST3GAL5, and ST8SIA1 in mp AD-MSCs, NI-mp AD-MSCs, or co-cultures with MSM. ACTB was used as a loading control.

**Figure 2 ijms-22-12194-f002:**
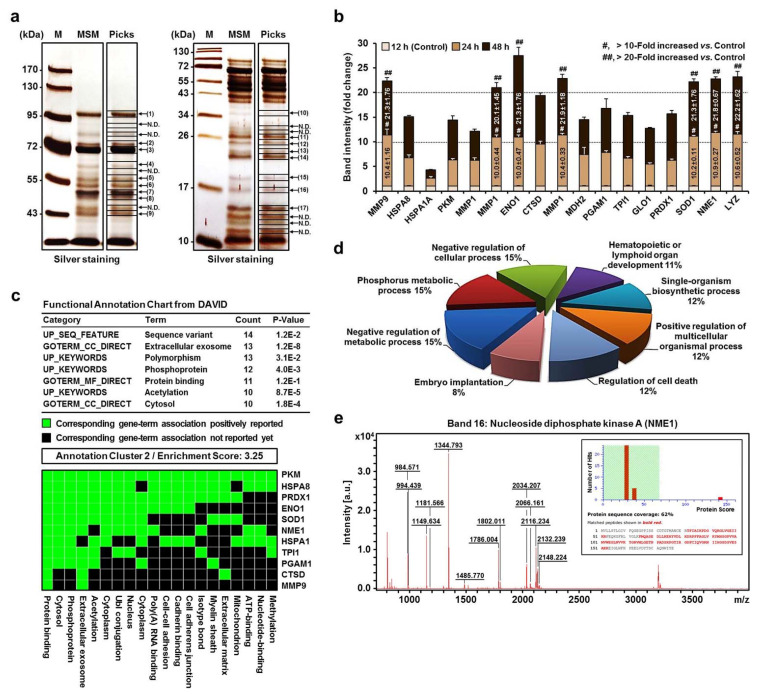
SDS-PAGE analysis and verification of high-abundance proteins in MSM. (**a**) SDS-PAGE and silver staining: pre-stained protein size marker (lane M), secretomes from cultured medium of active MPs (lane MSM), and sample picking from SDS-PAGE gel spots for peptide mass fingerprinting (PMF) analysis (lane Picks). N.D., not detected. (**b**) Quantification of protein bands from SDS-PAGE gels using Image J software. Data are presented as mean fold changes ± SD (*n* = 3; #, more than 10-fold increase versus the control, *p* < 0.05; ##, more than 20-fold increase versus the control, *p* < 0.05). (**c**) Two clusters were identified using the DAVID functional annotation clustering tool. Annotated cluster represents a kappa value > 0.35 and overlap = 3. Similarity scores ranged from high (>1) to low enrichment (<0.25). (**d**) Illustration of identified proteins analyzed by gene ontology tool. (**e**) PMF spectrum of NME1. Spectral masses (in mass per charge unit, *m*/*z*) obtained by MALDI-TOF MS were analyzed by bioinformatics using the Mascot and Profound search engines. The *x*-axis represents the mass-to-charge ratio (*m*/*z*), and the *y*-axis represents the relative abundance. The probability-based Mowse scores were obtained using the Mascot search engine. Among predicted proteins with differential Mowse scores shown as multiple bars on the *x*-axis, only proteins with Mowse scores > 68 were considered significant, which was 142 (*p* < 0.05) for NME1 protein. Red marks indicate peptides identified by mass fingerprinting that match calculated molecular weights. All protein identifications are provided in Appendix A.

**Figure 3 ijms-22-12194-f003:**
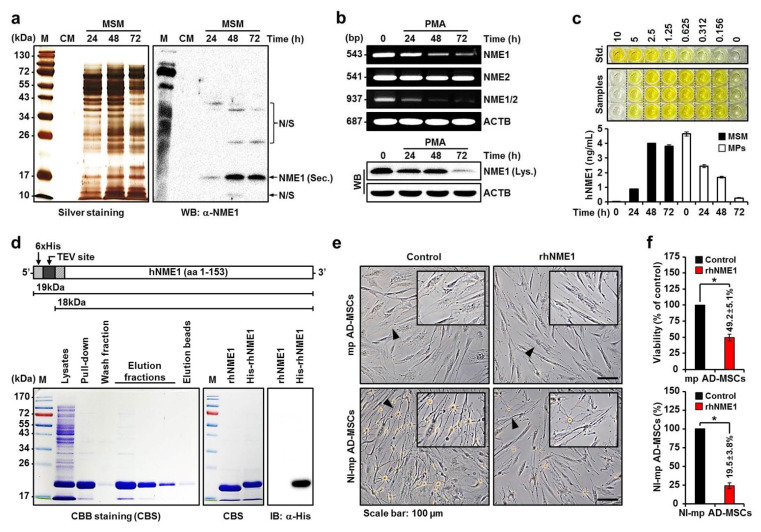
Quantitative analysis of secretome NME1 protein and functional evaluation of rhNME1. (**a**) MSM was prepared at various times, separated on SDS–PAGE, and subjected to silver staining or Western blotting using NME1 antibody. M, pre-stained protein size marker; CM, conditioned medium; N/S, non-specific; Sec., secretion; α-, anti-. (**b**) Reverse transcription polymerase chain reaction (RT-PCR) analysis of NME1, NME2, and NME1/2 mRNA in PMA-activated MPs (upper panel). Whole-cell lysates were prepared and subjected to Western blotting using NME1 antibody (lower panel). Lys., lysates. ACTB was used as a loading control. (**c**) ELISA results of MSM or MP intracellular hNME1. Standard color (top panel) and time-dependent color changes of samples (MSM and cells) (middle panel). Absorbance was recorded and quantified by conventional microplate readers (lower panel). The results shown are representative of three independent experiments. Std., standard (ng/mL). (**d**) The schematic structure and production of rhNME1 or His-rhNME1 proteins. The recombinant proteins consist of a 6 × His-tag and an extracellular domain of hNME1 with 153 amino acids (upper panel). The pET-28B (+)-hNME1 plasmids were transformed into *Escherichia coli* strain BL21 (DE3) and induced by isopropyl ß-D-1-thiogalactopyranoside. His-rhNME1 protein expression was analyzed by SDS-PAGE followed by Coomassie brilliant blue staining (lower left panel). The His-rhNME1 proteins were purified, subjected to SDS-PAGE followed by Coomassie brilliant blue staining (lower middle panel), and subjected to Western blot analysis with a His antibody (lower right panel). M, pre-stained protein size marker; CBS, Coomassie brilliant blue staining. (**e**) Phase-contrast images of cultured mp AD-MSCs and NI-mp AD-MSCs with or without rhNME1 (4 ng/mL). Arrowheads indicate enlarged images. (**f**) Cell viability was quantified by an MTS assay on mp AD-MSCs with or without rhNME1 (upper panel). The variations in the cell count of NI-mp AD-MSCs were quantified for co-cultures with or without rhNME1 (lower left panel). Data are presented as mean percentage levels ± SD (*n* = 3; * *p* < 0.05).

**Figure 4 ijms-22-12194-f004:**
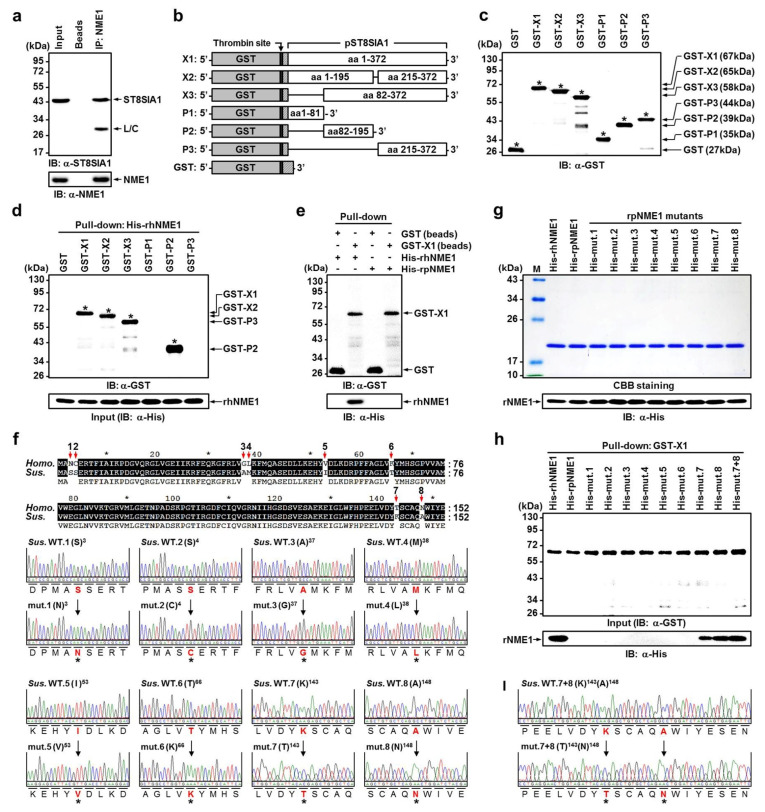
Identification of hNME1 as a novel pST8SIA1-binding protein using IP and pull-down assay. (**a**) NME1 interacts with ST8SIA1. U937 cells were lysed under non-denaturing conditions, and 500 μg of total lysate protein was subjected to IP with α-NME1. Precipitated proteins were resolved by SDS-PAGE, and IB analysis probed for ST8SIA1 or NME1 as indicated. Arrows indicate immunoreactive bands; the precipitated ST8SIA1 band is denoted with an asterisk. Beads, beads only (IgG); L/C, light chain. (**b**) The rpST8SIA1 constructs. GST fusion proteins were generated by cloning PCR-amplified ST8SIA1 sequences into the pEX-N-GST expression vector. Domain structure of the full-length GST-ST8SIA1 protein and the various deletion mutants used. X1-3, transcript variants 1–3; P1–3, partial transcripts 1–3; aa, amino acids. (**c**) Equivalent amounts of purified recombinant GST, GST-ST8SIA1, or GST-ST8SIA1 deletion mutants were analyzed by SDS-PAGE followed by IB analysis with the specific α-GST antibody. (**d**) GST pull-down assay on purified-His-rhNME1 incubated with equivalent amounts of GST, GST-X1, GST-X2, GST-X3, GST-P1, GST-P2, and GST-P3. Results were analyzed by SDS-PAGE followed by IB analysis with the specific α-GST and α-His antibody. The ST8SIA1 domains bound by NME1 are denoted with an asterisk. (**e**) GST pull-down assay on His-rhNME1 or purified-His-tagged recombinant porcine NME1 incubated with equivalent amounts of GST and GST-X1 beads. Results were analyzed by SDS-PAGE followed by IB analysis with the specific α-GST and α-His antibody, which showed that ST8SIA1 is attached only to hNME1 and not pNME1. (**f**) Comparison of the amino acid sequences of Homo sapiens (Homo.) and Sus scrofa (Sus.) NME1. The nucleotides Asparagine 3 (N3), Cysteine 4 (C4), Glycine 37 (G37), Leucine 38 (L38), Valine 53 (V53), Lysine 66 (K66), Threonine 143 (T143), and Asparagine 148 (N148) in the Homo. NME1 sequences were changed to Serine 3 (S3), Serine 4 (S4), Alanine 37 (A37), Methionine 38 (M38), Isoleucine 53 (I53), Threonine 66 (T66), Lysine 143 (K143), and Alanine 148 (A148) in the Sus. NME1 sequences, respectively. The DNA sequenced by Bionics (Bionics Co., Seoul, Korea) was analyzed using FinchTV 1.4.0 (Geospiza, Inc., Seattle, WA, USA). Sequence analysis results showed the Sus. NME1 cDNA and point-mutant NME1 cDNA in WT and mutants (mut.), respectively. (**g**) The expression of His-rhNME1, His-recombinant sus NME1 (rsNME1), and His-mut. 1–8 proteins were analyzed by SDS-PAGE followed by Coomassie brilliant blue staining and subjected to IB analysis with α-His antibody. (**h**) GST pull-down assay for His-rhNME1 or His-rsNME1, His-rsNME1, His-mut. 1–8, and His-mut. 7+8 incubated with equivalent amounts of GST-X1. Results were analyzed by SDS-PAGE followed by IB analysis with the specific α-GST and α-His antibody, which showed that rNME1 attached only to His-rhNME1, His-mut. 7, His-mut. 8, and His-mut. 7+8. (**i**) The amino acid sequence of His-mut. 7+8. The nucleotides 7. Threonine 143 (T143) and 8. Asparagine 148 (N148) in the Homo. NME1 sequences were changed to 7. Lysine 143 (K143) and 8. Alanine 148 (A148) in the Sus. NME1 sequences, respectively.

**Figure 5 ijms-22-12194-f005:**
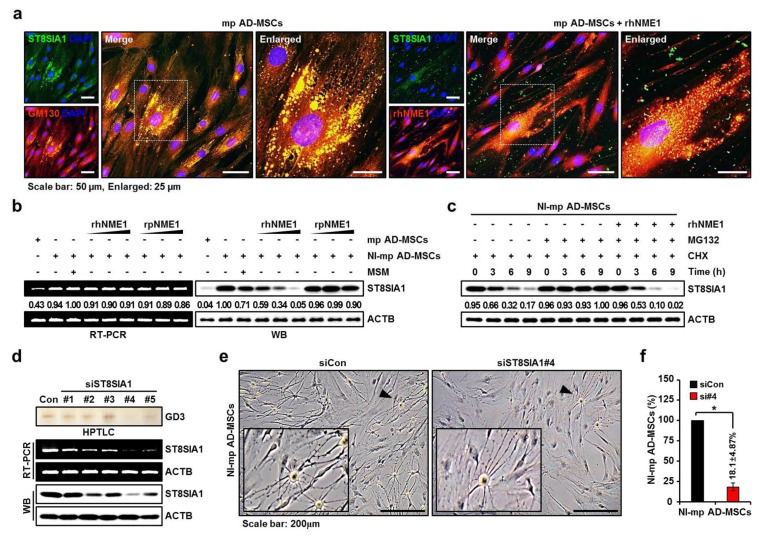
Immunocytochemistry of mp AD-MSCs with and without rhNME1 treatment. (**a**) Co-localization of endogenous ST8SIA1 in the Golgi apparatus in mp AD-MSCs or endogenous ST8SIA1 with exogenous rhNME1. Immunofluorescence microscopy results show mp AD-MSCs fixed and stained with antibodies against ST8SIA1 (green), cis-Golgi marker GM130 (red), and ectopic His-tagged rhNME1 (red). DAPI was used for nuclear counter staining. Overlapping staining (yellow) demonstrates the co-localization of the two proteins, and white squares indicate enlarged merged images. (**b**) Reduction in ST8SIA1 expression by rhNME1, rsNME1, and MSM treatment in mp AD-MSCs or NI-mp AD-MSCs. RT-PCR (left panel) and Western blots (right panel) for ST8SIA1 protein in rhNME1 or rsNME1-treated NI-mp AD-MSCs. Only rhNME1 (1, 2, and 4 ng/mL), not rsNME1, dose-dependently significantly decreased ST8SIA1 protein levels in NI-mp AD-MSCs; neither rhNME1 nor rsNME1 suppressed ST8SIA1 mRNA in NI-mp AD-MSCs. ACTB was used as a loading control. (**c**) ST8SIA1 is regulated by the proteasome and rhNME1. Western blot analysis determined the effects of rhNME1, MG132, and cycloheximide on ST8SIA1 protein accumulation. Protein extracts were prepared from NI-mp AD-MSCs treated with rhNME1 (4 ng/mL) or MG132 (10 uM), cycloheximide (150 nM), or a combination of the two. Equal protein loading was confirmed with ACTB antibody. (**d**) The knockdown of ST8SIA1 siRNA. ST8SIA1-specific siST8SIA1 or siCon was transfected into mp AD-MSCs. After transfection with siST8SIA1 (50 nM) for 48 h, ST8SIA1 expression was evaluated using HPTLC, RT-PCR, and Western blotting. ACTB was used as a loading control. All siRNA sequence information is provided in Appendix A. #, siRNA number. (**e**) Phase-contrast images of transfected NI-mp AD-MSCs with siCon or siST8SIA1 (#4). Arrowheads indicate enlarged images. (**f**) Variations in the cell count of NI-mp AD-MSCs were quantified for NI-mp AD-MSCs transfected with siCon or siST8SIA1#4 (si#4). Data are presented as mean percentage levels ± SD (*n* = 3; * *p* < 0.05).

**Figure 6 ijms-22-12194-f006:**
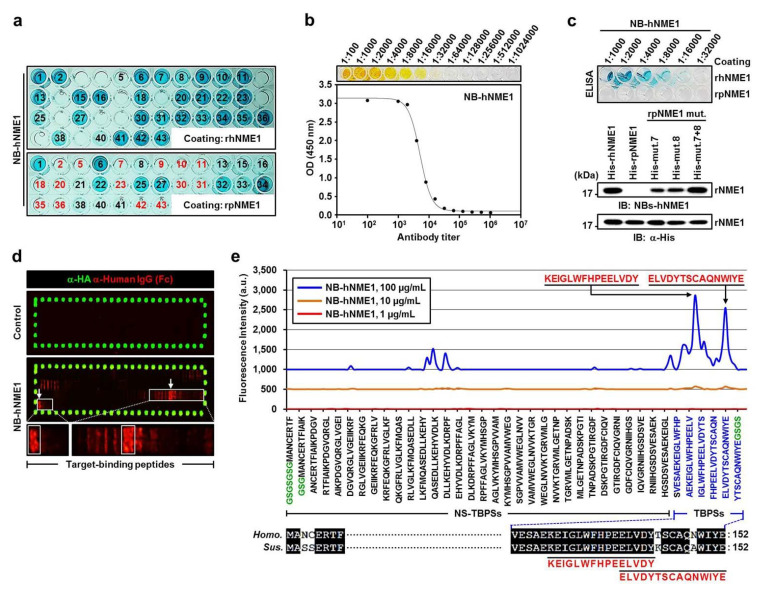
NB-hNME1 production and NME1 antigenic epitope mapping. (**a**) Detection of hNME1 antigen-specific antibodies by phage ELISA. Of 43 different colonies, 31 colonies (black number) were positive for rhNME1. Of these 31 rhNME1-positive colonies, 17 were used in the experiment; 15 colonies (red number) negative for rsNME1 were excluded. (**b**) Titer determination of NB-hNME1 by ELISA. The ratio of NB-hNME1 absorbance values at 450 nm. (**c**) The specificity of the hNME1 antigen in NB-hNME1 was confirmed by ELISA and IB analysis. Results were analyzed by SDS-PAGE followed by IB analysis with the NB-hNME1 and α-His antibody, which demonstrated that NB-hNME1 is attached only to His-rhNME1, His-mut. 7, His-mut. 8, and His-mut. 7+8. α-His antibody was used as a loading control. (**d**) The elongated antigen sequence was translated into 15 amino acid peptides with a peptide–peptide overlap of 14 amino acids. The resulting peptide microarrays contained 152 different peptides printed in duplicate (304 peptide spots) and were framed by additional HA (YPYDVPDYAG, 58 spots) control peptides. Goat α-human IgG (Fc) was used as secondary antibody; mouse monoclonal α-HA antibody was used as control. White squares indicate NB-hNME1-binding peptide; arrows indicate enlarged images. (**e**) Identification of a binding region of hNME1 interacting with NB-hNME1. A peptide array of hNME1 was exposed to NB-hNME1 at concentrations of 1, 10, and 100 µg/mL. The intensity plot of the peptide array signals appears as peaks with corresponding regions of hNME1. The two highest peaks are denoted with arrows in the graph, indicating two epitope-like spot patterns formed by adjacent peptides with the consensus motifs KEIGLWFHPEELVDY and ELVDYTSCAQNWIYE. GS (green text), neutral linkers at the C- and N-terminus; NS-TBPSs, nonspecific target-binding peptide sequences; TBPSs, target-binding peptide sequences.

**Figure 7 ijms-22-12194-f007:**
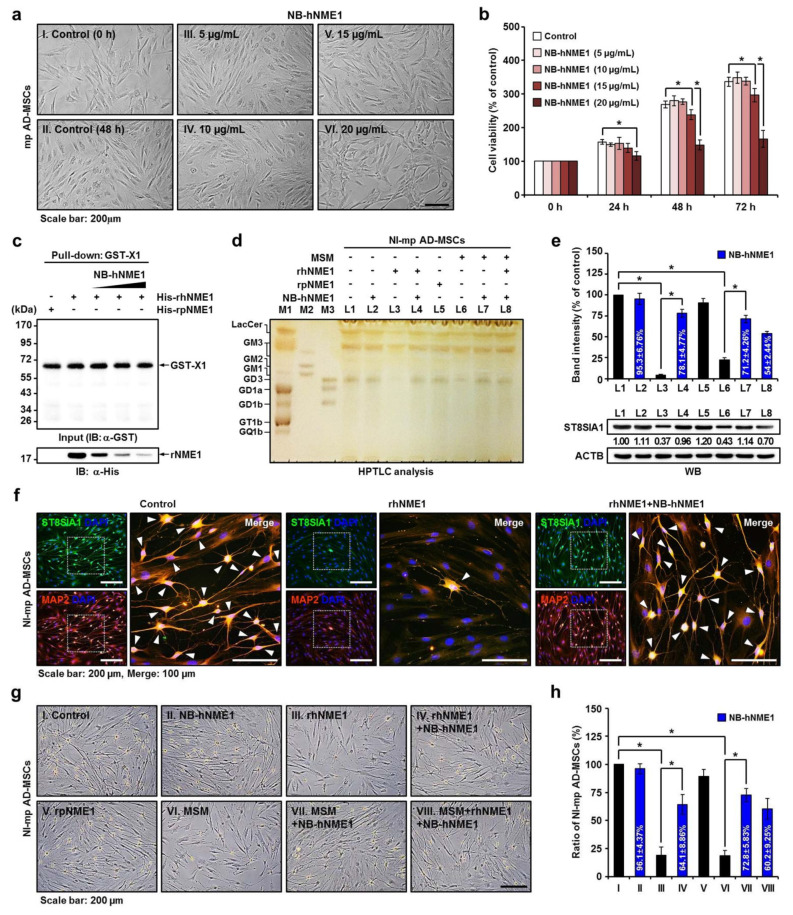
The effects of NB-hNME1 as an hNME1 suppressor in mp AD-MSC differentiation. (**a**) Determination of cytotoxicity and morphological changes in mp AD-MSCs after treatment with different doses (5, 10, 15, and 20 μg/mL) of NB-hNME1 for 48 h. (**b**) Cell viability was quantified by an MTS assay at 24, 48, and 72 h post-NB-hNME1 treatment. Data are presented as mean percentage levels ± SD (*n* = 3; * *p* < 0.05). (**c**) The effects of NB-hNME1 on GST-X1 protein and His-rhNME1 binding affinity. Results were analyzed by SDS-PAGE followed by IB analysis with the specific α-GST and α-His antibody, revealing that NB-hNME1 (5, 10, and 15 μg/mL) inhibits the interaction of hNME1 with pST8SIA1. (**d**) HPTLC of gangliosides in NI-mp AD-MSCs. Treatment with rhNME1 or MSM reduced ganglioside GD3 expression (L3 or L6), whereas co-treatment with rhNME1 or MSM and NB-hNME1 (L4 or L7) recovered ganglioside GD3 expression in NI-mp AD-MSCs. M1, adult mouse brain gangliosides; M2 and M3, ganglioside standard markers; L, line. (**e**) Quantification of band intensity (upper panel) and Western blots (lower panel) for ganglioside GD3 (L1 to L8). ACTB was used as a control. Data are presented as mean percentage levels ± SD (*n* = 3; * *p* < 0.05). L, line; L1, control; L2, NB-hNME1; L3, rhNME1; L4, rhNME1+NB-hNME1; L5, rsNME1; L6, MSM; L7, MSM+NB-hNME1; L8, MSM+rhNME1+NB-hNME1. (**f**) Immunofluorescence of NI-mp AD-MSCs (upper panel), NI-mp AD-MSCs cultured with rhNME1 (middle panel), or NI-mp AD-MSCs cultured with rhNME1 and NB-rhNME1 (lower panel). Immunofluorescence microscopy results show NI-mp AD-MSCs fixed and stained with antibodies against ST8SIA1 (green) and MAP2 (red). DAPI was used for nuclear counter staining. Overlapping staining (yellow) demonstrates the co-localization of the two proteins; white squares indicate enlarged merged images. (**g**) Phase-contrast images of cultured NI-mp AD-MSCs, NI-mp AD-MSCs with NB-hNME1, rhNME1, rsNME1, and MSM are shown. Neuronal differentiation of mp AD-MSCs was reduced by treatment with rhNME1 (III) or MSM (VI) and was recovered by co-treatment with rhNME1 or MSM and NB-hNME1 (IV. rhNME1+NB-hNME1; VII. MSM+NB-hNME1). (**h**) Variations in the cell count of NI-mp AD-MSCs were quantified for treatment with or without NB-hNME1, rhNME1, rsNME1, and MSM. Data are presented as mean percentage levels ± SD (*n* = 3; * *p* < 0.05). I, control; II, NB-hNME1; III, rhNME1; IV, rhNME+NB-hNME1; V, rpNME1; VI, MSM; VII, MSM+NB-hNME1; VIII, MSM+rhNME1+NB-hNME1.

## Data Availability

The data that support the findings of this study are available from the corresponding author Y.-K.C. (Young-Kug Choo) upon reasonable request.

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
