# Peer review of "The Potential Role of Human NME1 in Neuronal Differentiation of Porcine Mesenchymal Stem Cells: Application of NB-hNME1 as a Human NME1 Suppressor"

_ijms, 2021, doi:10.3390/ijms222212194_

Round 1

Reviewer 1 Report

The purpose of the manuscript entitled “The potential role of human NME1 in neuronal differentiation of porcine mesenchymal stem cells: application of NB-hNME1 as a human NME1 suppressor” was to investigate the effect of the human macrophage (MP) secretome on cell xenograft rejection.

The presentation of the work and the language used by the authors do not raise any major objections. The Introduction chapter provides an overview of the issues that are the subject of the paper, and the purpose of the paper is clearly stated, well justifying the research undertaken against the background of the existing state of knowledge. The methods used by the authors during the study are not only varied but, what should be emphasized, properly dedicated to solving the problems. The results obtained represent a new approach and have implications for the development of knowledge on the research topic addressed. However, some minor issues would need to be clarified. As a example: The question that generally arises is: wouldn't it be better to relate results from transfected cells to non-transfected controls? Also the question of how the cells are counted would need clarification. Some inconsistency is also apparent in the presentation of Fig7A-b results. The cells treated with 20 microg/ml seen in Fig7a do not appear to be in as bad of shape compared to the control as if it were in the graph (fig7b). The discussion is interesting, based on well-chosen scientific literature from recent years. Both the Introduction chapter containing the background of the conducted research and the Discussion of obtained results were described with reference to older works as well as to the newest literature concerning the described issues - 90 references were used, which are closely related to the subject of the work presented for review, citing them correctly.

Generally, the manuscript entitled “The potential role of human NME1 in neuronal differentiation of porcine mesenchymal stem cells: application of NB-hNME1 as a human NME1 suppressor” is interesting and seems acceptable with a little improvement.

Author Response

Point 1: Some inconsistency is also apparent in the presentation of Fig7A-b results. The cell treated with 20 μg/ml seen in Fig 7a do not appear to be in as bad of shape compared to the control as if it were in the graph (fig 7b).

Response 1: Thank you for providing these insights. You have raised an important point, however we wanted to determine the cell growth ratio of NB-hNME1 by concentration on mp AD-MSCs. In other word, Figure 7a-b in the paper show that it is not intended to confirm NB-hNME1 acts as a toxic agent and killing mp AD-MSCs. Therefore, a normal appearance in Figure 7a was observed compared to the control without changing the cell appearance, and because the growth was significantly lower than the control or the NB-hNME1 low concentration group, the concentration was judged unsuitable for experiment (Fig 7b).

Point 2: Also the question of how the cells are counted would need clarification.

Response 2: The reviewer was correct and we agree with your suggestion. (1) After we obtained images through a microscope, (2) a region with a constant cell distribution was randomly divided into 3 sections, (3) then we performed cell counting in manual. At this time, the target of cell counting is based on the nucleus of spindle-shaped cells expressing neuronal differentiation. (4) The relative values are displayed accordingly.

Reviewer 2 Report

In this manuscript Cho and Ju et al, evaluated the effect of macrophages (MPs) or macrophage secretion medium (MSM) on the proliferation and differentiation of miniature pig adipose tissue-derived mesenchymal stem cells (mp AD-MSCs), using co-cultures of mp AD-MSCs with MPs or MSM. Authors showed that the proliferation of mp AD-MSCs decreased drastically in the presence of MSM, but not with MPs, furthermore, MSM significantly reduced neuronal differentiation.  Further high-performance thin layer chromatography (HPTLC) authors reported that MSM reduced ganglioside GD3 expression in mp AD-MSCs and lead to decrease in neuronal differentiation.  Authors performed proteome analysis of MPs secretome and identified that hNME1 bind to porcine ST8 alpha-N-acetyl-neuraminide alpha-2,8-si- alyltransferase 1 (pST8SIA1), and induces degradation of pST8SIA1 in mp AD-MSCs and inhibit the ganglioside GD3 synthase, which ultimately results in decreased neuronal differentiation. Finally authors blocked the interaction between hNME1 and pST8SIA1 with nanaobodies NB-hNME1 and restored the neuronal differentiation of mp AD-MSCs. Based on these results authors proposed that NB-hNME1 can be used as an hNME1 suppressor for the neuronal differentiation of mp AD-MSCs, therefore it is a possible candidate for use as a additive such as an immunosuppressant, in xenogeneic stem cell transplantation.

This is an important study and the finding from this study have potential to use for neuronal differentiation through NB-hNME1.

This study cab be further improved by addressing the following minor comments:

In Figure S1 b authors showed two MSC markers CD90 and CD44, it will be better if authors also show the expression of CD73 and CD105.

Author Response

Point: In Figure S1b, authors showed two MSC markers CD90 and CD44, it will be better if authors also show the expression of CD73 and CD105.

Response: Thank you for providing these insights and in-depth analysis. We entirely agree with your suggestion and you have raised an important question. However, we believe that it is considered to be sufficient to prove the neuronal differentiation of MSCs such as confirming surface markers of MSC through Flow cytometry analysis, the morphological change of mp AD-MSCs, the gene expression of the neuronal differentiation including MAP2, NF-M, and GFAP through RT-PCR analysis.

Reviewer 3 Report

The manuscript by Cho and co-authors describes an interesting study evaluating the effect of the secretome of human macrophages in xenograft rejection.

The authors performed a lot of work including the establishment of a cell model of xenograft in vitro.

However, in this form, the manuscript is difficult to follow. 

The results section must be reorganized and the experiments presented in a more schematic form. The description of the results must be more coherent with the figures. For example, the comments in the results must be short and mostly moved in the discussion section.

Some conclusions, including the inhibition of neural differentiation, must be better shown. The immunofluorescences shown are poorly representative.

Figure 2 need to be better described and perhaps simplified. The same as the other figures.

The Introduction must be improved. Some aspects are missing ( eg. the role of Ganglioside GD3).

Author Response

Point 1: The results section must be reorganized and the experiments presented in a more schematic form. The description of the results must be more coherent with the figures. For example, the comments in the results must be short and mostly moved in the discussion section.

Response 1: Thank you for the thoughtful and precious feedback. We entirely agree with your suggestion and you have raised an important question. We have rewritten and relocated the general description of ganglioside (p. 2, lines 85-89) to be more in line with your comment, from Result 2.1 (p. 3, lines 130-134). We hope that the edited section clarifies the point we attempted to make. Also, the appropriate changes made in revised manuscript are highlighted.

Point 2: Figure 2 need to be better described and perhaps simplified. The same as the other figures.

Response 2: This is an interesting perspective. We agree that the description of figure should be better easy and make simplification to understand for potential readers, however we have retained some of our arguments as we found that the legend contains all the sufficient explanation for Figure 2. We eagerly hope you understand our rationale for this decision.

Point 3: The introduction must be improved. Some aspects are missing (eg. The role of Ganglioside GD3)

Response 3: Thank you for providing an important question and we agree with your assessment. The reviewer was correct: some aspects regarding the role of ganglioside GD3 were missing in Introduction. We have accordingly added the aspect on the role of ganglioside GD3 (p. 2, lines 89-92) and these are also highlighted.

Round 2

Reviewer 3 Report

no comments